# Research Progress of Nature-Inspired Metaheuristic Algorithms in Mobile Robot Path Planning

**Yiqi Xu, Qiongqiong Li, Xuan Xu, Jiafu Yang *** and **Yong Chen**

College of Mechanical and Electronic Engineering, Nanjing Forestry University, Nanjing 210037, China;
xuyiqi@njfu.edu.cn (Y.X.)
* Correspondence: jfyang@njfu.edu.cn; Tel.: +86-139-5100-4006

**Abstract:** The research of mobile robot path planning has shifted from the static environment to the dynamic environment, from the two-dimensional environment to the high-dimensional environment, and from the single-robot system to the multi-robot system. As the core technology for mobile robots to realize autonomous positioning and navigation, path-planning technology should plan collision-free and smooth paths for mobile robots in obstructed environments, which requires path-planning algorithms with a certain degree of intelligence. Metaheuristic algorithms are widely used in various optimization problems due to their algorithmic intelligence, and they have become the most effective algorithm to solve complex optimization problems in the field of mobile robot path planning. Based on a comprehensive analysis of existing path-planning algorithms, this paper proposes a new algorithm classification. Based on this classification, we focus on the firefly algorithm (FA) and the cuckoo search algorithm (CS), complemented by the dragonfly algorithm (DA), the whale optimization algorithm (WOA), and the sparrow search algorithm (SSA). During the analysis of the above algorithms, this paper summarizes the current research results of mobile robot path planning and proposes the future development trend of mobile robot path planning.

**Keywords:** mobile robot; path planning; metaheuristic algorithm; firefly algorithm; cuckoo search algorithm

## 1. Introduction

The term "mobile robot" refers to a machine system that can move autonomously or partially autonomously in a variety of environments. It is supposed to possess the ability to perceive the surrounding environment through highly sensitive sensors, accurately identify and comprehend its position, create precise maps of the environment, enable safe navigation to the intended destination, and accomplish pre-determined tasks [1,2]. Mobile robots can be categorized into three groups based on their operating environments [3]: (i) ground mobile robots, such as autonomous vehicles (AV) and autonomous guided vehicles (AGV); (ii) marine mobile robots [4–6], such as autonomous underwater vehicles (AUV) and surface mobile vehicles (SMV); (iii) air mobile robots, such as unmanned aerial vehicles (UAV). Table 1 shows the categorization of mobile robots according to the operating environments and their main planning tasks in different operating environments. Mobile robots are now utilized in the applications of industry, agriculture, military, medicine, and entertainment because of the developments in computational intelligence and sensor accuracy. However, regardless of the application, mobile robots need to have a certain level of autonomy in a working environment without human intervention [7–9].

The implementation of autonomous control technology for mobile robots depends on four aspects: motion, perception, cognition, and navigation [10]. The structure of the main components that make up a mobile robot is depicted in Figure 1. Navigation technology has extensive development prospects since it is essential for mobile robots to achieve autonomous movement and adaptive planning. Navigation technology [11] includes four

basic requirements: perception, localization, cognition, and path planning, among which path planning is the most fundamental research task [12–14].

**Table 1.** Planning tasks and applications of mobile robots in different environments.

| Classification of Mobile Robots (Operating Conditions) | | Major Planning Tasks | Mission |
|---|---|---|---|
| Ground mobile robots | Autonomous vehicles (AV) Autonomous guided vehicles (AGV) | 1. avoiding obstacles; 2. shortest route planning; 3. dynamic environmental adaptation; 4. multi-objective path planning. | auto-navigation; transportation of materials; patrol (police, army, or navy); search and rescue missions. |
| Marine mobile robots | Underwater mobile vehicles (UMV) Surface mobile vehicles (SMV) | 1. avoiding obstacles; 2. adaptation to the marine environment; 3. safety of navigation; 4. optimization of energy; 5. interaction with other vessels. | exploration for marine resources; seabed topographic mapping; marine ecological monitoring. |
| Air mobile robots | Unmanned aerial vehicles (UAV) | 1. avoiding obstacles; 2. air route planning (ARP); 3. optimization of energy; 4. considering wind resistance and meteorological factors; 5. avoiding air traffic. | battlefield reconnaissance; electronic reconnaissance; mine detection; laser guidance. |

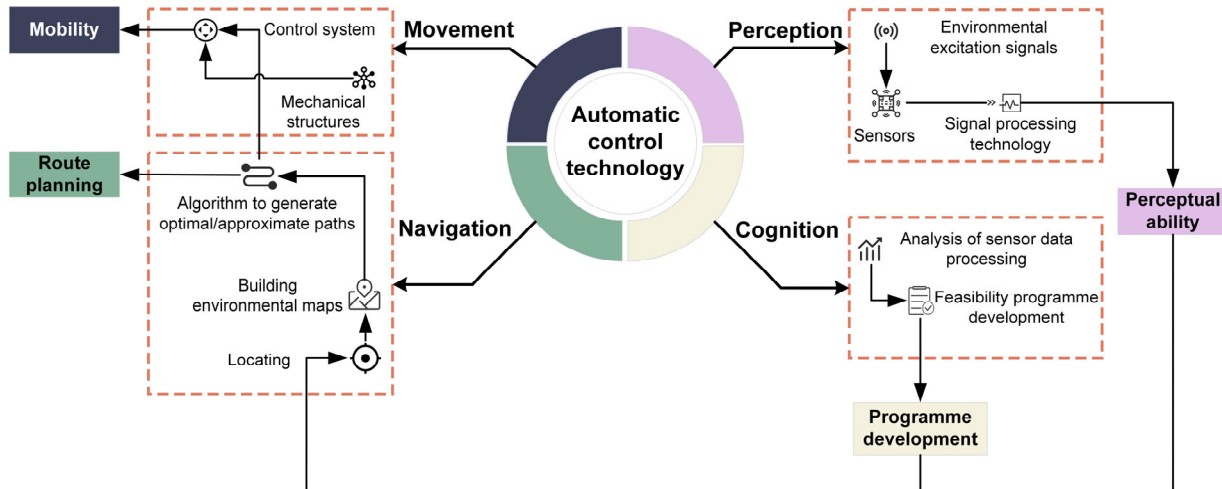

**Figure 1.** Structural diagram of the key technology in mobile robots.

Path planning for mobile robots belongs to a class of "Nondeterministic Polynomial"(NP) problems [15]. Its solution is usually guessable and verifiable in polynomial time. However, at present, a universally applicable method for solving NP problems has yet to be discovered, resulting in inherent uncertainty when dealing with this topic [16,17]. In most cases, there are multiple feasible paths in the search space from the starting position to the target position, and the decision of which path to take as the optimal or approximate solution to the problem is determined by some guiding criteria (e.g., shortest distance, path smoothness, minimum energy consumption, etc.) [18,19]. As a result, path-planning algorithms have grown to be a significant issue in the field of mobile robots [20,21]. As the configuration region dimension increases, so does the complexity, and some classical path-planning algorithms that were once widely used no longer suffice [22]. At present, path-planning algorithms are divided into two types: heuristic algorithms and metaheuristic algorithms. Heuristic algorithms are a class of algorithms tailored to solve a specific problem, which are capable of solving problems within a reasonable timeframe or finding approximate solutions in cases where traditional methods fail to provide precise solutions.

However, similar to most classical methods, heuristic algorithms still need to be based on a specific problem framework, which restricts their broad application [23]. In recent years, metaheuristic algorithms have evolved to become capable of addressing a wide range of optimization problems without changing the core algorithmic foundation. Regardless of the structure or feature of a challenge, they are capable of finding workable solutions [24]. As a result, metaheuristic algorithms are widely regarded as effective approaches for solving a wide range of optimization problems and have become popular algorithms for addressing practical optimization problems in a variety of fields [25–33]. Table 2 summarizes the relevant characterization of path-planning algorithms for different classes of mobile robots.

**Table 2.** The table of the performance of different kinds of path-planning algorithms.

| Types of Algorithms | Typical Algorithms | Advantages | Limitations | Versatility |
|---|---|---|---|---|
| traditional algorithms | CD | efficient at finding the shortest paths | struggles with complex obstacle distributions | applicable to finding the shortest paths in various environments |
| | PRM | applicable to various types of maps and obstacle distributions | requires pre-building of the graph, not suitable for dynamic environments | approximate optimal solutions |
| | RRT | efficient in high-dimensional space | may generate non-smooth paths | high-dimensional space path planning |
| heuristic algorithms | A* | heuristic search is more efficient | requires appropriate heuristic function design | single-source shortest-path problems |
| | Dijkstra | simple and easy to implement | high-time complexity (inefficient on dense graphs) | applicable to non-negative weighted graphs |
| metaheuristic algorithms | PSO | strong global optimization ability, fast convergence | requires appropriate parameter settings | applicable in path planning for obstacle avoidance and global optimization problems |
| | FA | fast convergence, strong global search ability | may require longer search time for complex problems | applicable to global optimization problems |
| | CS | fast convergence and performs well in complex problems | requires appropriate parameter settings | applicable to global optimization problems |

There has been tremendous progress in the application of metaheuristics to mobile robot path-planning problems over the last two decades [34–37]. In this discipline, the genetic algorithm (GA), ant colony optimization (ACO), and particle swarm optimization (PSO) are the most well-studied and representative algorithms. Numerous studies [38–53] have been conducted on these algorithms, which are regarded as being typical for mobile robot path planning. Recently, researchers have shown increasing interest in newly developed algorithms such as the firefly algorithm (FA) and the cuckoo Search algorithm (CS) [35–37,54,55].

In Section 2, we propose a novel classification method based on nature, human, and discipline behavior. In Section 3, we evaluate the research development of nature behavior algorithms in mobile robot path planning by using the FA algorithm and the CS algorithm as the key algorithms. In Section 4, we mainly discuss the research progress of the FA algorithm and the CS algorithm in mobile robot path planning, along with the current development status of metaheuristic algorithms along with the urgent problems in the future development of metaheuristic algorithms. In Section 5, we summarize the above and put forward new ideas for future research in mobile robot path planning.

## 2. Metaheuristic Algorithms

The expression "metaheuristic algorithms" refers to a family of algorithms inspired by human intelligence or nature, which can be broadly classified as a type of stochastic

optimization algorithm. When information is lacking or knowledge of the problem under consideration is insufficient, metaheuristic algorithms are thought to be a class of optimization techniques that are independent of the problem [56]. As a result, they can be widely applied to the majority of optimization problems, as well as highly nonlinear and discrete problems [57,58].

### 2.1. The Fundamental Principle of Metaheuristic Algorithms

Metaheuristic algorithms can be considered a general algorithmic framework or a black-box optimizer. They can be applied to practically every optimization problem since they make few presumptions about the issues. In the vast majority of cases, metaheuristic algorithms typically outperform heuristic algorithms in optimization situations [59,60]. Laporte and Osman described the mechanisms of metaheuristic algorithms in [61] as "intelligently combining different concepts, mechanisms for generating iterative lower-level heuristics through guidance and constructing information within learning strategies, exploring and exploiting the search space in a series of steps to more efficiently find solutions that are close to optimal". Similarly, ref. [62] defined the mechanism of metaheuristic algorithms as "a process of exploration and exploitation". The terms "exploitation" and "exploration" summarize the search mechanisms of metaheuristic algorithms graphically. "Exploration" is the ability of the algorithm to search the environment comprehensively and generate different solutions; "exploitation" is the ability of the algorithm to search the local area and find the current optimal or suboptimal solution. The idea of randomization in metaheuristic algorithms enables the algorithm to transition from local searching to global searching. Therefore, almost all metaheuristic algorithms are suitable for the issue of global optimization problems. The "exploration" capability adds diversity to solutions, whereas the "exploitation" capability adds intensity to solutions. While the intensity of local search prevents slow algorithmic convergence and boosts the quality of solutions, the randomization in global exploration helps avoid becoming stuck in local optima and raises the diversity of solutions. As a result, the harmony between these two talents determines how well metaheuristic algorithms perform in their search for global optimal solutions. Nowadays, metaheuristic algorithms are viewed as a group of search techniques that include developing heuristics, carrying out local searches, and more general guidelines for resolving particular issues [63].

### 2.2. The Development and Classification of Metaheuristic Algorithms

GA [64] is a type of metaheuristic algorithm inspired by the theory of "Darwinian evolution". It is widely regarded as the first metaheuristic algorithm because it simulates the evolutionary process to search for optimal solutions. As the research progressed, scholars traced back to earlier studies on metaheuristic algorithms, such as the simulated annealing algorithm (SA) [65] invented by Kirkpatrick et al. in 1983. In 1986, the Tabu Search algorithm (TS) [66] was proposed, and in 1995, Kennedy and Eberhart proposed PSO [67]. In 1999, Dorigo et al. established an ant colony model and proposed ACO [68]. Figure 2 illustrates the development of metaheuristic algorithms. Since the beginning of the 21st century, there has been explosive growth in the development of metaheuristic algorithms, with new algorithms being proposed almost every year. According to incomplete statistics, there are hundreds of metaheuristic algorithms and their variants [56,57,69,70] at present.

For metaheuristic algorithms, there are currently three common categorization methods: (1) based on the source of inspiration [24,71–74]; (2) based on whether they are nature-inspired [24,73,75]; (3) based on individual or trajectory [61,73,76,77]. Here is the translation of the three classification methods for metaheuristic algorithms: The first classification method, based on the source of inspiration, divides metaheuristic algorithms into swarm intelligence algorithms, evolutionary algorithms, bio-inspired algorithms, and natural science algorithms. The second classification method, based on whether they are naturally inspired, categorizes metaheuristic algorithms into natural inspiration algorithms and non-natural inspiration algorithms. The third classification method divides metaheuris-

tic algorithms into individual-based algorithms and trajectory-based algorithms. Relying only on method 1 might result in algorithm intersections, where an algorithm (such as PSO) may belong to both the biological inspiration and the swarm intelligence inspiration. Simply relying on method 2 or method 3 cannot comprehensively cover the majority of metaheuristic algorithms. By studying the development of metaheuristic algorithms and the inspirations of classical metaheuristic algorithms [57,60,62,71,78], we find that the ideas of metaheuristic algorithms are derived from the natural behavior of animals and plants, human social behavior, and scientific behavior. Therefore, this paper combines methods 1 and 2 and proposes a new classification method based on natural behavior, human behavior, and scientific behavior. Based on this proposed classification method, Figure 3 presents an incomplete classification of the mainstream metaheuristic algorithm. Table 3 summarizes the advantages and disadvantages of the algorithms under the classification method proposed in this paper.

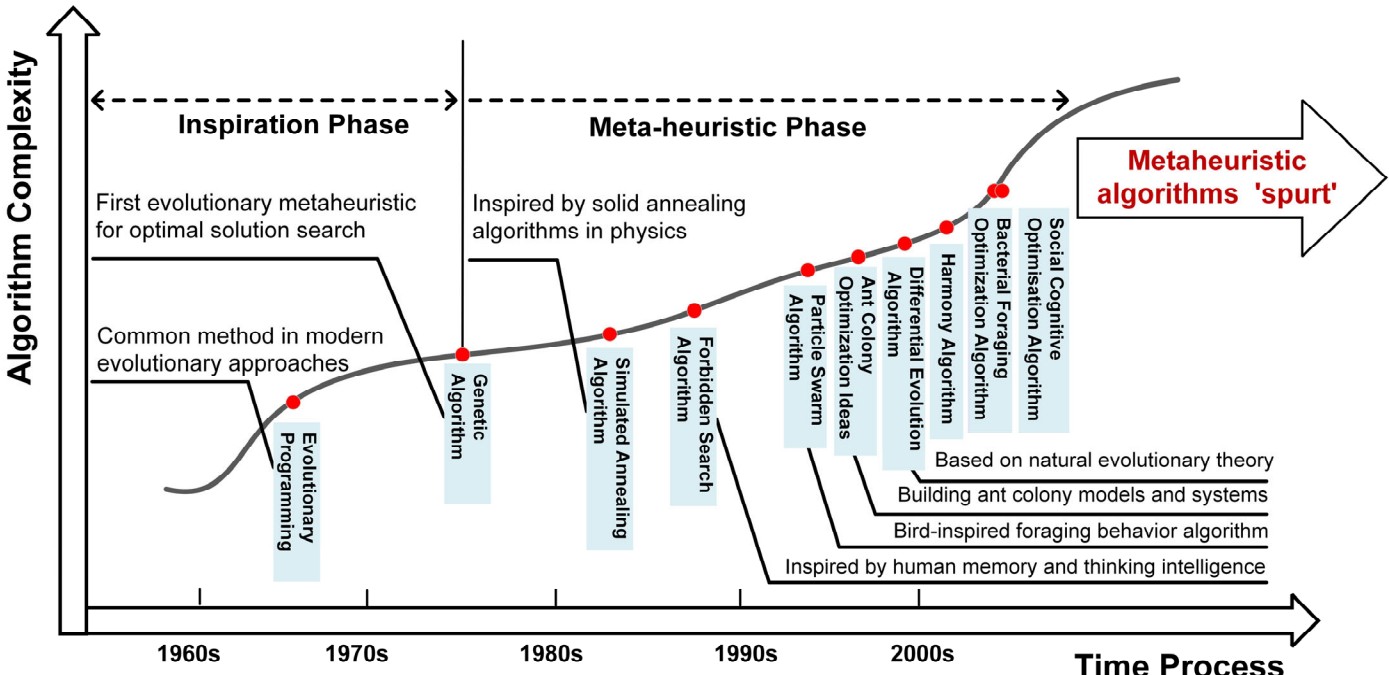

**Figure 2.** The development diagram of metaheuristic algorithms.

**Table 3.** The advantages and disadvantages of typical metaheuristic algorithms.

| Classification Method | Typical Algorithms | Advantages | Disadvantages |
|---|---|---|---|
| Natural behavior-based | PSO GWO FA CS | 1. simple structure and principles; 2. intelligent and robust; 3. adaptive organization; 4. balanced global and local search capabilities. | 1. long iteration time; 2. artificial parameter pettings. |
| Human social behavior-based | IWD ICA | 1. stronger global search capability; 2. fewer parameter settings. | 1. lack of diversity of viable solutions; 2. artificial parameter settings. |
| Discipline behavior-based | GWA BHA | 1. strong localized search capability; 2. small size of calculated costs. | 1. complex algorithmic principles; 2. weak global search capability; 3. artificial parameter settings. |

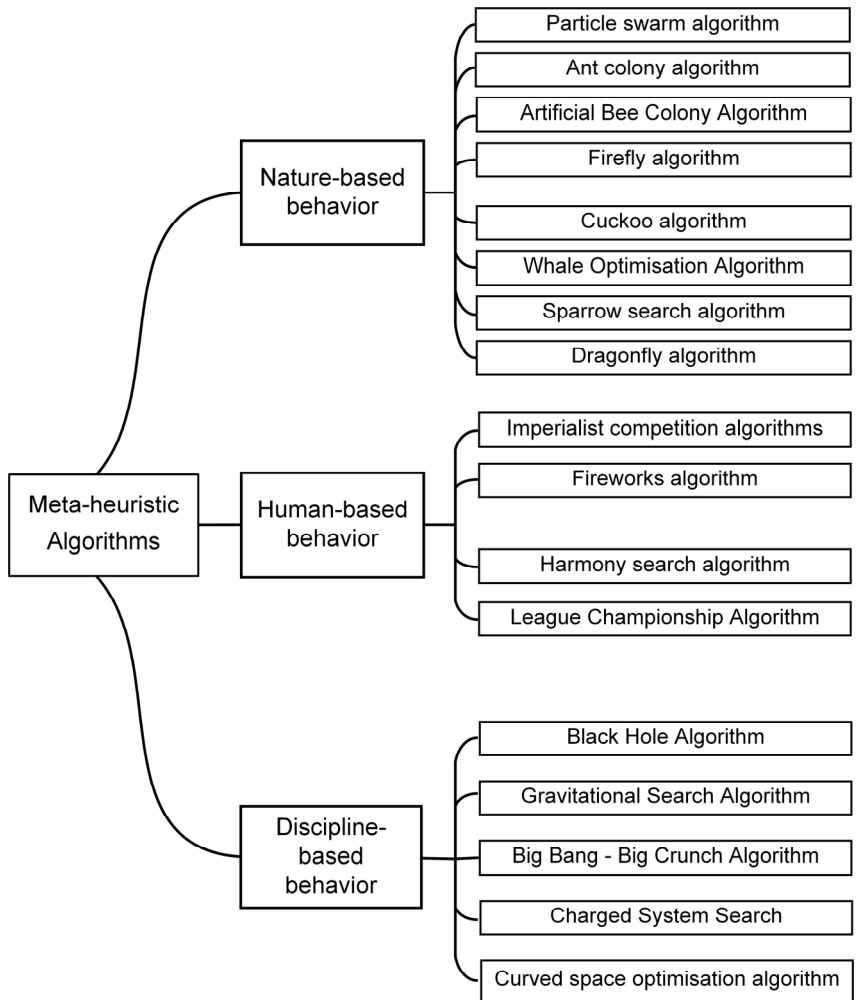

**Figure 3.** The classification chart of metaheuristic algorithms.

Among the three categories, the metaheuristic algorithm based on natural behavior demonstrates an exceptional ability to strike a balance between global search and local search, making it the most prevalent choice in mobile robot path planning. However, it does come with the drawback of needing to traverse all feasible solutions during a complete iterative search, resulting in slower convergence speed and increased memory consumption.

### 2.3. Nature-Inspired Metaheuristic Algorithms

Table 4 displays an incomplete compilation of classical and novel nature-inspired metaheuristic algorithms that are often used in mobile robot path planning since the 1990s. PSO and ACO are the two metaheuristic algorithms inspired by natural behavior that are most used in mobile robot path planning with the most abundant variants due to their early years of proposal. With the increasing scale and complexity of mobile robot path planning, recent years have witnessed a rise in interest among academics in intelligent algorithms such as the artificial bee colony algorithm (ABC), FA, CS, DA, WOA, SSA, and other algorithms, of which the most widely studied algorithms are the FA and DA, and in addition, the most promising algorithms are the DA, WOA, and SSA.

**Table 4.** Metaheuristic algorithms based on natural behavior.

| Complete Name of the Algorithm | Abbreviation | Year of Invention | Main Application Environments |
|---|---|---|---|
| Particle swarm optimization [79–82] | PSO | 1995 | Static\dynamic\multi-robot |
| Ant colony optimization [39–42,44,83–86] | ACO | 1991 | Static\dynamic\multi-robot |
| Bacterial foraging algorithm [87] | BFA | 2002 | Static |
| Artificial bee colony algorithm [88–90] | ABC | 2005 | Static\dynamic\multi-robot |
| Grey wolf optimizer algorithm [91–98] | GWO | 2007 | Muti-robot |
| Firefly algorithm [99–101] | FA | 2009 | Dynamic\multi-dimensional |
| Cuckoo search algorithm [102–104] | CS | 2009 | Dynamic\multi-dimensional\multi-robot |
| Dragonfly algorithm [105] | DA | 2015 | Dynamic\multi-robot\heterogeneous system |
| Whale optimization algorithm [106] | WOA | 2016 | Dynamic\unknown\multi-robot |
| Squirrel search algorithm [107] | SSA | 2020 | Dynamic\unknown\multi-robot |

## 3. Progress of Nature-Based Behavior Algorithms in Mobile Robot Path Planning

The schematic diagram of the FA and CS algorithm applied in path planning reviewed in this paper is shown in Figure 4. As Figure 4 shows, in mobile robot path planning, $(X_S, Y_S)$ represents the starting point, while $(X_G, Y_G)$ represents the ending point, and dynamic obstacles are represented by yellow matrix blocks. Ideally, there are no obstacles at the start and end points and the global map environment is known. The mobile robot follows the principle of the shortest straight-line distance between two points for global planning (indicated by the gray dashed line). When the robot moves to a certain position, the sensor recognizes a dynamic obstacle at the position $(X_C, Y_C)$. At this time, the mobile robot needs to avoid dynamic obstacles and perform local path planning to re-plan an optimal path to the endpoint. As the algorithms search for the optimal solution with different mechanisms, the location of the next waypoint is determined differently when performing local planning, resulting in multiple feasible path points and multiple feasible paths. In Figure 4, we have listed some example points, such as points $(X_C', Y_C')$, $(X_C'', Y_C'')$, $(X_C''', Y_C''')$. The path generated by the path-planning algorithm is usually required to be the minimization of one or more objective optimization functions [108]. Therefore, at this stage, the path-planning problem, from simply planning a collision-free smooth route from the starting point to the ending point in different environments, gradually evolves into a class of multi-objective optimization problems and implements local path planning in dynamic environments to find the global optimal solution or approximate solution [37]. In this section, we take the FA and CS as the primary algorithms to analyze and study the present research state of metaheuristic algorithms based on natural behaviors in mobile robot path planning.

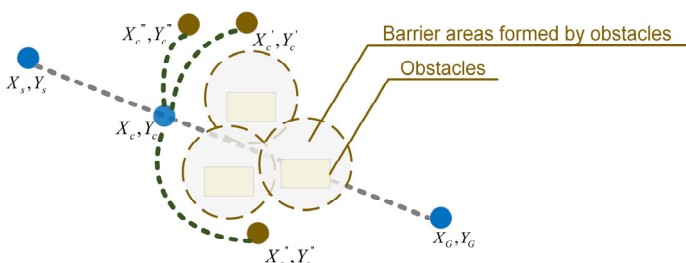

**Figure 4.** Schematic diagram of mobile robot path-planning problems.

### 3.1. Firefly Algorithm

The FA is a metaheuristic algorithm proposed by Yang XS [109], inspired by the behavior of fireflies in nature, where they emit light to attract mates or ward off enemies. Three rational properties for the FA were introduced by Yang XS: (1) All individual fireflies are gender-neutral, and theoretically, any two fireflies have an attraction or can be attracted to a relationship. (2) The brightness of fireflies and the distance between them are directly inversely correlated with how appealing they are. The lower-brightness firefly will travel toward the higher-brightness one if there are two flashing fireflies. A particular firefly will move arbitrarily if there is not another firefly brighter than it. (3) The cost function (light intensity), which needs to be tuned, determines the luminosity of fireflies. Figure 5 shows the flowchart and phases of the FA. Moreover, Algorithm 1 shows the pseudo-code diagram of the FA.

---

**Algorithm 1**: The Pseudo-code diagram of the FA.

---

Input:
Population size(n)
Maximum number of iterations (max_iterations)
Attraction coefficient (beta0)
Absorption coefficient (gamma)
Lower bounds of variables (lb)
Upper bounds of variables (ub)
Objective function to be optimized (f)
**While** ($t <$ max_*iterations*),
for $i = 1 : n$ (*population size*($n$))
   for $j = 1 : n$ (*population size*($n$))
   if ($I_i < I_j$)
     Initialize fireflies[i][j] randomly between lb[j] and ub[j]
   end if
    Evaluate new solutions and update light intensity.
   end for j
end for i
Rank the fireflies and find the current global best position
end while
Output:
The best solution found (the firefly with the highest brightness)

---

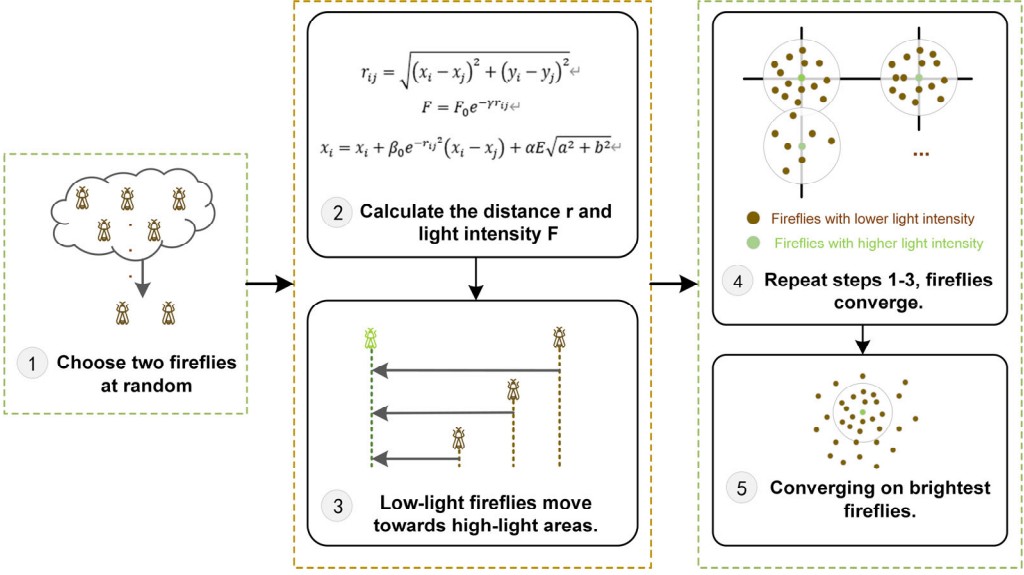

**Figure 5.** Schematic diagram of the FA.

Since its inception, it has been demonstrated that the FA outperforms PSO and the GA when it comes to searching for optimal solutions for specific complex optimization problems. Because of its simplicity, the limited number of algorithm parameters, and simplicity in implementation, the FA is efficient for tackling complicated multi-objective NP problems. Although the FA outperforms other algorithms in terms of optimization speed and solution correctness, it still has several problems that need to be fixed. For instance, the FA is prone to becoming trapped in local optima, its performance significantly depends on the choice of control parameters, and it prematurely converges during implementation [110–114]. Researchers have modified the algorithm from several perspectives to address these problems, which may be divided into three primary categories: (1) improved attraction mode strategies, (2) adaptive parameter control strategies, and (3) hybrid improvement strategies.

There have been several FA variations since 2016. The significant research advancements are listed in Figure 6 below.

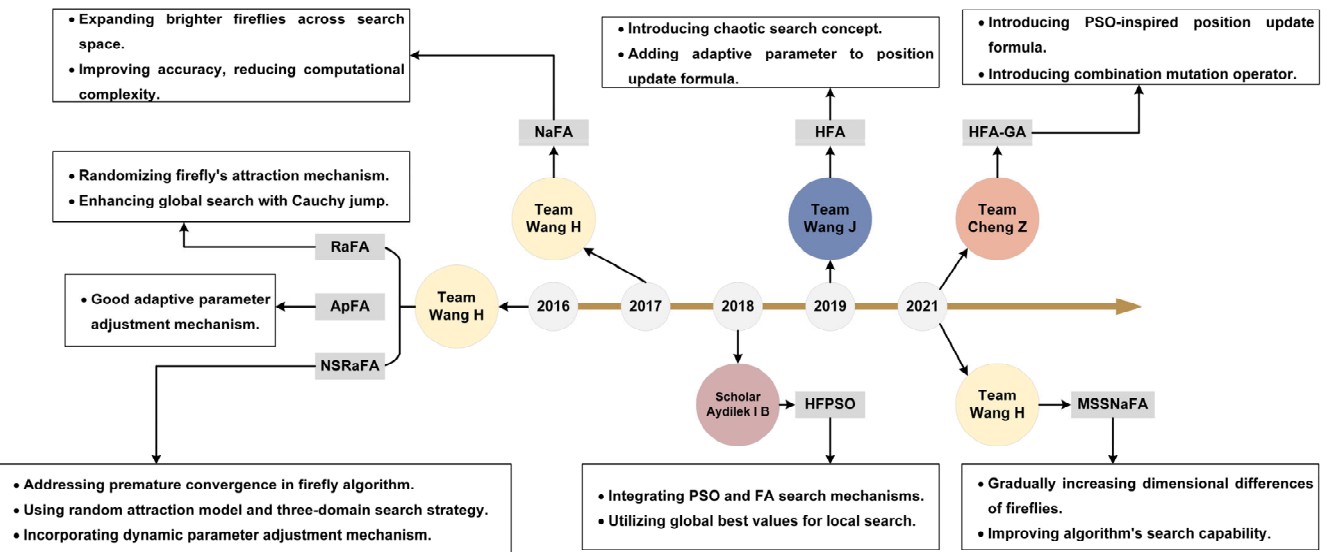

**Figure 6.** The significant research advancements of the FA.

Regarding the mobile robot path-planning problem, before 2018, mobile robot path-planning research was limited to static situations. In [115], the MO-FA algorithm with innovative evolutionary operators was suggested. It is designed to address three goals: optimal path length, smoothness, and safety of the way. The algorithm was statistically assessed and evaluated using multi-objective metrics such as HV and SC through simulation experiments in eight different static real-world scenarios. The outcomes demonstrated that the modified algorithm performed more effectively than the widely used NSGA-II strategy. In [116], an improved FA called the path center-based computation method FA (PPMFA) was proposed. It addressed the issue of convergence in the standard algorithm by replacing the fixed step size search mechanism with a Gaussian random walk, which diversified the population and improved the algorithm's random search capability. The algorithm increased the success rate of generating optimal solutions when used in conjunction with a dual-checking strategy. Comparative investigations revealed that in terms of accuracy and convergence speed, the proposed algorithm performed superior to PSO and the standard FA. In [110], a hybrid optimization algorithm called the improved whale-firefly optimization (IWFO) was proposed to overcome the shortcomings of the standard FA in multi-robot path planning, such as being prone to local optima or premature convergence. The IWFO introduced IWO into the standard FA to accelerate convergence, allowing for more accurate feasible solutions with less computational time.

However, investigations increasingly expanded to dynamic environments after 2018, with a focus on hybrid methods and parameter adaptability optimization. In [117], ACO was introduced into the standard FA, resulting in a hybrid algorithm called the hybrid

ACO-FA (HAFA). The results of the experiments demonstrated that this modified algorithm could locate approximation optimal solutions at a faster convergence rate. According to [118], an improved FA algorithm named the modified FA (mFA) was presented to fulfill goals including path smoothness, path safety, and low computational time cost. It added a directional random approach and enhanced the mobile step size in the standard FA. The results of simulation experiments validated the efficiency of this modified strategy. Additionally, this optimization strategy provided an innovative approach for resolving path-planning issues for mobile robots in environments with three-dimensional spherical obstacles that are known, partially known, or unknown. In the same year, reference [119] focused on autonomous planning for AUVs and considered runtime constraints, energy consumption issues, and the uncertainty of dynamic unknown underwater environments. They proposed a differential evolution-based FA optimization (DEFO). The improved strategy was tested in a complex three-dimensional simulation environment to validate its adaptive performance. Through Monte Carlo trials, the effectiveness of the hybrid algorithm's model was demonstrated, showing low computational costs and a certain level of real-time capability. Flinders University has used the model for extensive experimentation and study.

In 2019, a random guided firefly algorithm (ERaFA) based on elite strategy has been proposed in [113] to tackle challenges such as high computational time complexity and sluggish convergence speed in complicated environments. The algorithm introduced a crossover operator mechanism based on the GA to enhance its local search ability. It proved through verification utilizing the CEC2015 benchmark set and three limited engineering problems that the ERaFA performed superior in terms of convergence speed performance than the standard FA, RaFA, and ApFA. In [112], a modified FA named the modified FA (MFA) was proposed. By controlling the parameters, the algorithm's optimization ability was enhanced. Experimental results showed that when the MFA was used for path planning, the robot's trajectory length was shorter compared to the standard GA and FA, thus improving the efficiency of mobile robot operations. This laid the foundation for research on mobile robot path planning in unknown environments.

In 2020, to address the issue of the standard FA frequently becoming trapped in local optima and improve its performance, ref. [120] suggested an improved adaptive dynamic fuzzy planning FA algorithm (ADFA) by introducing standard fuzzy rules. A dynamic adaptive FA algorithm (GDAFA) was proposed in [114]. The global movement mechanism in this improved algorithm facilitated the dynamic modification of step size and attractiveness. The overall evolutionary optimization efficiency of the strategy was enhanced by combining a Gaussian distribution with an adaptive deviation strategy based on optimal distance. The GDAFA demonstrated significant advantages in terms of convergence time as well as solution correctness through validation utilizing 18 distinct optimization characteristics of classical test functions and engineering constraint issues.

In 2023, a hybrid FA-TPM strategy for complicated dynamic situations was proposed in [121]. This new system employed an adaptive selection strategy based on changes in the environment to allow mobile robots to swiftly determine the most effective collision-free path. The effectiveness of the algorithm was confirmed using real-world validation with a robot powered by an Atmel ATMEGA8 central CPU and OpenGL simulation on a software platform.

From 2015 to 2022, FA algorithms have achieved significant results in the field of mobile robot path planning. Table 5 summarizes some of the key research results.

**Table 5.** The significant research progress of the FA in mobile robot path planning.

| Year | Research Focus | Examples of Research |
|---|---|---|
| Before 2018 | Optimal path search in multi-objective optimization requirements | [110] [115] [116] |
| Before 2018 | Multi-robot coordination in complex static environments | [115] [116] |
| After 2019 | Extension to dynamic and high-dimensional environments | [112] [118] [119] [121] |
| After 2020 | Advancements in parameter optimization and hybrid intelligence algorithms | [112] [118] [119] [120] |
| After 2023 | Advancements in parameter optimization and hybrid intelligence algorithms | [121] |

*3.2. Cuckoo Search Algorithm*

The cuckoo, a bird found in nature, possesses a beautiful song, but along with its melodious tunes, it exhibits aggressive behavior in its brood parasitism. Cuckoos can mimic the color and pattern of host bird eggs, allowing them to deceive other birds by laying their eggs in the host birds' nests. Host birds have two ways of dealing with parasitic cuckoo eggs: either abandoning the eggs and the nest or incubating the cuckoo eggs. If the eggs are successfully hatched, the cuckoo chicks will push the host's eggs out of the nest during their growth process. Yang and Deb presented the CS in 2008 [122], a metaheuristic algorithm that was motivated by the cuckoos' natural parasitic activity. As shown in Figure 7, the CS developed by Yang and Deb simulates the parasitic behavior of cuckoos and follows three idealized criteria: (1) a cuckoo can only lay one egg at a time and randomly place it in a host nest; (2) nests with high-quality eggs (solutions) are preserved for the next generation; (3) the probability of a host bird discovering a parasitic egg is denoted as $P_a$ (where $P_a \in (0,1)$). The objective of these three criteria is to achieve an optimal solution search in the CS, implementing the idea of the algorithm: replacing less promising solutions with potentially better ones [123]. The CS is based on the Levy flight criterion, which differs from the random step size mechanism. In Levy flight, the step size follows a heavy tailed distribution, enabling the algorithm to explore feasible solutions more effectively [124]. Algorithm 2 shows the pseudo-code diagram of the CS.

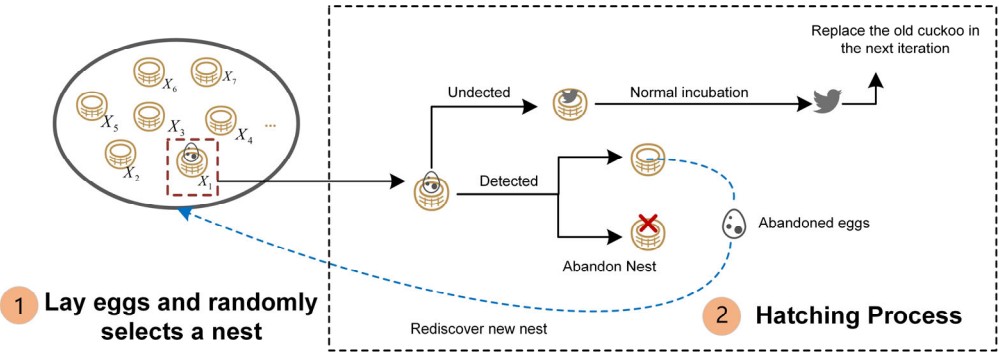

**Figure 7.** Schematic diagram of the CS algorithm.

---

**Algorithm 2:** The Pseudo-code diagram of the CS

---

Input:
Population size (n)
Maximum number of iterations (max_iterations)
Cuckoo egg laying rate (pa)
Step size scaling factor (alpha)
Lower bounds of variables (lb)
Upper bounds of variables (ub)
Objective function to be optimized (f)
**While** ($t <$ max_$iterations$),
for $i = 1 : n$ $(population\ size(n))$
　　for $j = 1 : n$ $(population\ size(n))$
　　if $(I_i < I_j)$
　　　Initialize fireflies[i][j] randomly between lb[j] and ub[j]
　　end if
　　　Evaluate new solutions and update light intensity.
　　end for j
end for i
Rank the cuckoo and find the current global best position
end while
Output:
The best solution found (the cuckoo with the highest fitness)

---

Since the introduction of the CS, it has been widely applied in various engineering problems, including image processing, robotics, biotechnology, and predictive modeling, due to its minimal initialization parameters and simple principles, making it suitable for multi-objective optimization problems. Ref. [103] compiled and categorized 12 typical variants of the CS and their respective application domains from its inception to 2017. Ref. [125] provided a concise summary of the development of the variants of the CS, which can be broadly categorized into two groups: (1) directly optimizing the basic parameters of the step size and (2) indirectly optimizing the basic parameters of the step size. The first group either introduces a dynamic adaptive approach using the conventional parameters or replaces the Levy distribution with another distribution form, such as the Gaussian or Cauchy distribution. The second group develops a new mathematical function model to regulate the parameters for updating the position at each iteration, combining the benefits of various algorithms with the CS. In addition, a fresh version of the CS (NMS-CS) was put forth in [125], and 23 traditional benchmark functions are used to demonstrate its effectiveness. Additionally, utilizing three engineering design issues, the accuracy of the proposed variant strategy was compared with the PSO, GSA, and GWO algorithms, demonstrating the effectiveness of the NMS-CS in addressing multi-constraint optimization problems.

For the mobile robot path-planning problems, in 2015, a hybrid optimization algorithm for navigation of multiple mobile robots (CS-ANFIS) based on the CS and least squares estimation (LSE) was proposed in [126]. The ANFIS parameters were optimized using the CS and LSE, and the mobile robot's turning angle was determined to prevent robot collisions. Real mobile robots were employed to verify the strategy, while simulation experiments were carried out in a static environment. The results demonstrated that the proposed strategy is adaptable to any difficult environment. It was recommended that dynamic obstacles be taken into account in future advancements in mobile robot technology.

In 2018, a dynamic adaptive CS algorithm (ACS) was put forth in [127] to address the issue of safe navigation of mobile robots in chaotic environments. This dynamic strategy identified the most optimal target direction based on the fitness function value by first evaluating the mobile robot's position regarding the goal and obstacles. Ref. [127] was the initial attempt to use the dynamic adaptive CS algorithm in mobile robot path planning, serving as the basis for further study. In the same year, ref. [128] provided an improved

strategy for the CS-bat hybrid algorithm. Ref. [128] provided a new strategy for optimizing algorithms by combining two or more metaheuristic algorithms. However, the hybrid algorithm presented in [128] focused on path planning in static environments, and future research in this field should expand to complex dynamic environments.

In 2019, research on the CS in mobile robot path planning mainly focused on the extension and application of the dynamic adaptive strategy proposed in [127] and the hybrid intelligent algorithm strategy proposed in [128]. In [129], the challenge of spatially optimal path planning for moving robots was discussed, and a hybrid genetic cuckoo algorithm was suggested. This hybrid strategy eliminated the reliance on algorithm parameters by introducing the crossover and mutation processes of the genetic algorithm into the population initialization phase of the standard CS. The weaknesses of a single algorithm were made up for by the hybrid algorithm. Based on the results of the simulation, it was demonstrated that the hybrid algorithm could reconcile the conflict between computing time and an optimal solution. In [130], an improved CS algorithm (chaotic CS) was applied to UCAV path planning. To improve the dynamic flexibility of the initial parameters throughout the iteration process as well as the search effectiveness of the algorithm in locating global optimal solutions, the circular chaotic mapping concept was added to the standard CS. Through trials on six benchmark functions, the performance of the chaotic CS was examined. This showed how flexible the hybrid algorithm is when it comes to optimizing complex and multimodal objective functions. For the three-dimensional path planning of mobile robots, ref. [102] proposed an improved CS algorithm based on compact parallel technology. The concept of a compact algorithm was introduced into the standard CS using a probability model to represent the entire population. With less storage consumption, this method produced a performance that was comparable to the standard algorithm. To further enhance the performance of the compact algorithm, a parallel strategy was introduced. The proposed parallel CCS algorithm outperformed the ACS proposed in [127] on 16 benchmark functions, but it exhibited poor performance on certain benchmark functions (f7, f9, f10, etc.).

Based on [127], ref. [131], published in 2020, provided an intelligent SCS algorithm and developed a new fitness function. To address the weak convergence issue of the standard CS, they created an adaptive dynamic tuning parameter mechanism. According to simulation results, the SCS can adapt to various environments and navigate smoothly and safely to their destination. In the same year, ref. [132] presented an improved CS algorithm based on a competition selection function. Instead of using the random selection idea included in the standard CS, this algorithm replaced it with a competition selection function that calculated the best route for robots from their initial position to their final position. Instead of using the random selection idea included in the standard CS, this algorithm replaced it with a competition selection function that calculated the best route for robots from their initial position to their final position. The introduction of the competition selection function mechanism overcame the problem of robots becoming stuck in local minima while searching in the search space, increasing the probability of finding the optimal result. When testing this improved algorithm in a multi-robot system, experimental results demonstrated a 5–8% improvement in computational time and path length.

In 2021, based on the multi-robot path planning proposed in [126], ref. [104] presented a dual-robot global path-planning algorithm based on the CS. Through simulations, the algorithm was shown to possess better global planning capability.

In 2022, ref. [133] presented a cuckoo-beetle hybrid algorithm (CBSS), combining beetle populations into the cuckoo population to address the path-planning problems of heterogeneous robots. The standard CS frequently runs into problems including premature convergence and local minima. The proposed CBSS was applied to two-dimensional and three-dimensional path planning for heterogeneous robots. In comparison to earlier proposed similar algorithms, the CBSS could guarantee locating the shortest global optimal path on maps of various dimensions and types.

From 2018 to 2022, CS algorithms have achieved significant results in the field of mobile robot path planning. Table 6 summarizes some of the key research results.

**Table 6.** The significant research progress of the CS in mobile robot path planning.

| Year | Research Milestones and Algorithms | Literature References |
|---|---|---|
| 2015 | Initial research on CS-ANFIS for multi-mobile robot navigation and optimization | [126] |
| 2016 | Continued focus on CS-ANFIS as a hybrid intelligent algorithm for mobile robot path planning | — |
| 2017 | CS-ANFIS remained a significant research focus in mobile robot path planning | — |
| 2018 | Introduction of dynamic adaptive optimization CS algorithm (ACS) for unstructured mobile robot path planning; proposal of CS-bat algorithm | [127] [128] |
| 2019 | The emergence of several algorithms for optimal path planning in multi-dimensional spaces: hybrid genetic-cuckoo algorithm; chaotic CS algorithm; parallel CCS algorithm; intelligent SCS algorithm | [129] [130] [102] |
| 2020 | Continued research on algorithms for mobile robot path planning in multi-dimensional spaces | [131] [132] |
| 2021 | The research focus shifted to collaborative path-planning for multi-mobile robots, especially heterogeneous robots, in high-dimensional spatial environments | [104] [133] |

### 3.3. Other Algorithms

Based on the systematic application summary of the FA and CS in mobile robot path-planning problems in Sections 3.1 and 3.2, these two intelligent algorithms have made contributions to various path-planning problems for mobile robots, including path planning in complex or dynamic environments, multi-objective path planning, and collaborative path planning of multiple robots in 3D spatial environments. The application of nature-inspired heuristic algorithms in mobile robot path planning is not limited to the FA and CS. In 2015, Mirjalili developed the dragonfly algorithm (DA) [105] by studying the natural behavior of dragonflies in evading predators and finding food. The following year, Mirjalili et al. proposed the whale optimization algorithm (WOA) [134], inspired by the hunting behavior of whales. In 2020, Jian Shuang Cui from Donghua University in China introduced the sparrow search algorithm (SSA) [135] by simulating the foraging and anti-predation behavior of sparrows in nature. These three algorithms were proposed relatively recently, and research on their application in mobile robot path planning is still relatively scattered, lacking a comprehensive research framework. However, over the past two years, these three algorithms have made some significant research advances in the fields of real-time path planning for heterogeneous multi-robot systems in complex multi-dimensional environments, as well as single/multi-mobile robot path-planning problems in dynamic unknown environments or complex multi-dimensional environments. They have paved the way for more study in this area and offered innovative viewpoints.

A similar article [136] in the same year addressed real-time path planning for hybrid UAV/UGV systems in unknown 3D environments. They proposed a grid-based path-planning strategy that combined biomimetic neural networks with an optimized DA algorithm. The initial search process of the DA algorithm was modified to increase search efficiency by developing a three-dimensional dynamic motion model based on bio-inspired neural networks. Simulation experiments demonstrated that the improved dynamic pro-

gramming algorithm effectively accomplished real-time trajectory planning tasks for the hybrid UAV/UGV systems in different states, including static and dynamic environments, static and dynamic targets, and more.

In 2021, ref. [137] proposed a fuzzy-WOA optimization strategy by fusing the WOA with fuzzy control technology. The proposed innovative approach demonstrated a 20.63% increase in path length through simulation trials that took into account static and dynamic settings as well as single/multi-robot real-time scenarios. In [138], the exploration problem of mobile robots in unknown environments was addressed, which also developed the FMH-WOA integration strategy in a multi-robot detection system. This algorithm combined deterministic CME technology with the standard WOA. According to simulations, the strategy demonstrated effectiveness in all situations by adding a frequency modification function to modify the random parameters in the WOA. Ref. [139] provided an improved ISSA strategy on robot route planning in actual environments based on local route smoothing (LPS), a neighborhood search approach, and an improved position update formula. The algorithm's advantages of quick convergence and robust optimization in path-planning issues have been verified by simulation studies that proved its effectiveness. It was additionally stated that the modified algorithm might be used in the future for dynamic obstacle avoidance and multi-robot path planning. Simulation experiments verified the algorithm's effectiveness and verified its advantages of rapid convergence and robust optimization in path-planning problems. It was additionally suggested that the modified algorithm might be used in the future for dynamic obstacle avoidance and multi-robot path planning. An improved fusion strategy for the problem of multiple drone path planning in complicated hilly situations was put forth in [140]. The SSA, improved BINN (biogeography-based neural network) technology, and B-spline curve technology were all combined in this algorithm. The path-planning problem for numerous drones in complicated mountainous landscapes was successfully solved by the fusion algorithm through simulation trials and analysis, revealing considerable benefits in terms of safety and path length. In [107], the path-planning problem for mobile robots in unexplored environments was discussed, and an improved SSA strategy was suggested. The approach developed a hybrid fitness function taking into account path length and safety and introduced the fitness-distance-balance (FDB) mechanism and the Harris Hawks algorithm (HHA) as inspiration. Through CEC2017 experiments, the effectiveness of the suggested modified ISSA has been verified.

In 2022, ref. [106] aimed to solve the path-planning problem for mobile robots in highly complex dynamic environments. An improved whale optimization algorithm (NWOA) was proposed to address issues such as slow convergence speed and lack of dynamic obstacle avoidance capability in the standard WOA and its optimization variants. The NWOA incorporated an improved potential field factor into the standard WOA to enhance the robot's dynamic obstacle avoidance ability. Simulation experiments showed that the proposed NWOA exhibited faster convergence speed and improved dynamic planning performance in mobile robot path planning.

## 4. Discussion

(1) At present, the FA and CS have formed a preliminary framework in the research of path planning in complex and multidimensional environments. The future research direction is to further study optimization in algorithm performance based on the existing optimization techniques with the fusion algorithm and adaptive parameter improvement as the main optimization strategies. Based on the current application research of the FA in dynamic environments, future research should aim at the improvement and optimization of the FA in unknown environments and establish a perfect parameter optimization mechanism to achieve high robustness and high adaptive performance of the algorithm in unknown environments. The CS shows excellent performance in dynamic and high-dimensional environments, but most of the algorithms are still tested in simulation platforms, and a standardized and reasonable evaluation system

has not been established to evaluate the proposed improvement algorithms, which makes the proposed improvement algorithms not universal and generalizable. Future research should establish a standardized mathematical evaluation mechanism to verify the rationality of the optimization algorithm, and at the same time, the algorithm effectiveness test should break through the virtual environment established by the simulation platform and be extended to the real scenario for physical testing. The DA, WOA, and SSA are cutting-edge research in the field of mobile robot path planning, especially in dealing with unknown space and heterogeneous multi-robot systems. However, it is not possible to establish a framework for the optimization of these three algorithms. Table 7 summarizes the algorithmic complexity and computer hardware requirements of the FA, CS, DA, SSA, and WOA algorithms to guide subsequent research.

(2)  The focus of future mobile robot path-planning research tends to be (1) solving optimization problems in path planning of single or multi-robot systems in complex dynamic environments with low computational costs; (2) solving safety and smoothness in path planning of spatial robots or heterogeneous robot systems in unknown multidimensional environments, combined with the ultimate development goal of mobile robots to replace humans in unknown and dangerous environments The ultimate development goal of mobile robots is to achieve fully autonomous exploration tasks in unknown hazardous environments instead of humans. Therefore, the path-planning algorithm research should continue to study in depth the two optimization strategies of dynamic adaptive optimization of parameters and fusion of intelligent algorithms, in addition to the combination of general artificial intelligence (GAI) techniques, such as (AI-generated content, AIGC). The future research direction is to consider the dynamic parameter adaptive optimization strategy by combining various hyperparametric optimal configuration strategies (HPO), such as resampling error estimation based on supervised machine learning for adaptive parameter modification, which will boost the intelligence of these three algorithms in mobile robot path planning.

(3)  For multi-objective optimization NP problems such as path planning for mobile robots, the current technology cannot find one or more algorithms to solve such problems. The intelligent algorithms reviewed in this paper have shown some intelligence and effectiveness in dealing with complex optimization problems, but according to the "No Free Lunch Theorem" (NFL), it is difficult to find a general and effective algorithm for solving all optimization problems. In particular, in the field of mobile robot path planning, it is impossible to find one or more metaheuristic algorithms that can adapt to all environmental states or meet all practical problem requirements, so only suitable algorithms can be selected according to actual application scenarios or desired goals. Metaheuristic algorithms are one of the effective algorithms for solving multi-objective optimization problems, and new metaheuristic algorithms are proposed every year. However, almost all metaheuristic algorithms suffer from the problem of imbalance between global and local search ability during the global optimal solution search, mainly because a complete mathematical analysis theory has not been established to evaluate the performance of metaheuristic algorithms. The current research mainly relies on various evaluation mechanisms to subjectively verify the effectiveness of the algorithms, which lacks objectivity. Future research should be devoted to developing a sound objective mathematical evaluation mechanism to further improve the balance between global search and local search of metaheuristic algorithms, thus enhancing the solution quality.

**Table 7.** The analysis of practical applications of algorithms.

| Algorithms | Computing Resources | Computational Complexity | Computational Time | Requirements for Onboard Vehicle Computers | Recommended Computer Configuration |
|---|---|---|---|---|---|
| FA | Moderate to high CPU and RAM | Moderate | Moderate | Adequate CPU and RAM for algorithm execution, suitable for vehicles with moderate computing capabilities | CPU: 4 core clock speed of 2.5 GHz or high; RAM: 8 GB; Storage: 128 GB |
| CS | Moderate CPU and RAM | Moderate | Moderate | Adequate CPU and RAM for algorithm execution, suitable for vehicles with moderate computing capabilities | CPU: 4 core clock speed of 2.5 GHz or high; RAM: 8 GB; Storage: 128 GB |
| WOA | Moderate to high CPU and RAM | Moderate | Moderate | Adequate CPU and RAM for algorithm execution, suitable for vehicles with moderate computing capabilities | CPU: 4 core clock speed of 2.5 GHz or high; RAM: 8 GB; Storage: 128 GB |
| SSA | Low CPU and RAM | Low | Low | Low CPU and RAM requirements, suitable for vehicles with limited computing capabilities | CPU: 2 core clock speed of 2.5 GHz or high; RAM: 4 GB; Storage: 256 GB |
| DA | Moderate CPU and RAM | Moderate | Moderate | Reasonably capable CPU and RAM for algorithm execution, suitable for vehicles with moderate computing capabilities | CPU: 4 core clock speed of 2.5 GHz or high; RAM: 8 GB; Storage: 128 GB |

## 5. Conclusions

In summary, research in the field of mobile robot path planning has made some progress, and the FA and CS have shown excellent performance in complex scenarios. The DA, WOA, and SSA, on the other hand, have the potential for cutting-edge research. However, to further advance the research, physical testing of the effectiveness of the algorithms should be enhanced and combined with GAI techniques to develop intelligent path-planning algorithms for mobile robot systems.

**Author Contributions:** Conceptualization, Y.X.; methodology, Y.X.; software, Y.X.; validation, Y.X.; formal analysis, Y.X.; investigation, Q.L.; resources, J.Y.; data curation, X.X.; writing—original draft preparation, Y.X.; writing—review and editing, J.Y.; visualization, Q.L; supervision, J.Y.; project administration, Y.C.; funding acquisition, J.Y. All authors have read and agreed to the published version of the manuscript.

**Funding:** This research was funded by the National Natural Science Foundation of China (Grant No. 32072498).

**Data Availability Statement:** Not applicable.

**Conflicts of Interest:** The authors declare no conflict of interest.

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
