# Peer review of "Research Progress of Nature-Inspired Metaheuristic Algorithms in Mobile Robot Path Planning"

_electronics, doi:10.3390/electronics12153263_

Round 1
Reviewer 1 Report
The authors should address the following comments to improve the quality of paper for publication in Electronics journal.
11. The paper's presentation and organization need improvement. It is currently too text-heavy and may appear tedious to readers. Consider including more tables, flowcharts, and descriptive figures to enhance readability. Additionally, proofreading is highly recommended.
22. The contributions of the paper are unclear and insufficiently integrated into the manuscript. At the end of the Introduction section, it is essential for the authors to clearly highlight the paper's contributions, emphasizing their strength and distinct significance compared to the state-of-the-art.
33. The authors should provide a more critical review of recent studies in the state-of-the-art, particularly focusing on the path planning domain. Here are some examples of relevant references:
– Yazdani, Amir Mehdi, Karl Sammut, O. Yakimenko, and Andrew Lammas. "A survey of underwater docking guidance systems." Robotics and Autonomous Systems 124 (2020): 103382.
– Qin, H., Shao, S., Wang, T., Yu, X., Jiang, Y. and Cao, Z., 2023. Review of autonomous path planning algorithms for mobile robots. Drones, 7(3), p.211.
– MahmoudZadeh, S., Abbasi, A., Yazdani, A., Wang, H. and Liu, Y., 2022. Uninterrupted path planning system for Multi-USV sampling mission in a cluttered ocean environment. Ocean engineering, 254, p.111328.
– Liu, L., Wang, X., Yang, X., Liu, H., Li, J. and Wang, P., 2023. Path planning techniques for mobile robots: Review and prospect. Expert Systems with Applications, p.120254.
– Abbasi, A., MahmoudZadeh, S., Yazdani, A. and Moshayedi, A.J., 2022. Feasibility assessment of Kian-I mobile robot for autonomous navigation. Neural Computing and Applications, pp.1-20.
– Hewawasam, H.S., Ibrahim, M.Y. and Appuhamillage, G.K., 2022. Past, present and future of path-planning algorithms for mobile robot navigation in dynamic environments. IEEE Open Journal of the Industrial Electronics Society, 3, pp.353-365.
4. The quality of the figures needs improvement, particularly the clarity of labels used in the figures.
5. Authors should establish a concrete correlation between the proposed metaheuristic algorithms and the context of path planning. For instance, explain how the proposed methods can be applied to "Global Path Planning," "Local Path Planning," "Obstacle Avoidance," "Re-Planning," etc.
6. Authors should provide a thorough discussion of the computational complexity and practical applicability of the proposed methods in path planning applications.
7. Authors should critically discuss different path planning/optimization algorithms, preferably in the form of a table (e.g., Table 1), including additional attributes for comparison such as Advantages, Limitations, Versatility, etc. Gain insights from the following reference: https://doi.org/10.1016/j.robot.2019.103382
8. Authors should provide a brief pseudocode of the main proposed algorithms used in this paper. Gain insights from the following reference: https://doi.org/10.1016/j.eswa.2023.120254
9. The Conclusion section should be revised and written in a standard format. The paper's contributions (in such a format) should be explicitly addressed in the Introduction.
Proofreading is highly recommended.
Author Response
Response to Reviewer 1 Comments
Point 1: The paper's presentation and organization need improvement. It is currently too text-heavy and may appear tedious to readers. Consider including more tables, flowcharts, and descriptive figures to enhance readability. Additionally, proofreading is highly recommended.
Response 1: We completely agree with your point. To enhance the readability of the paper, we have incorporated several tables and images. For instance, Table 1, Table 2, Table 3, Table 5, Table 6, Table 7, and Figure 7 have been added.
Point 2: The contributions of the paper are unclear and insufficiently integrated into the manuscript. At the end of the Introduction section, it is essential for the authors to highlight the paper's contributions, emphasizing their strength and distinct significance compared to the state-of-the-art.
Response 2: We completely agree with your point. We have made the following improvements to lines 90-97 in the Introduction section.
Point 3: The authors should provide a more critical review of recent studies in the state-of-the-art, particularly focusing on the path planning domain. Here are some examples of relevant references.
Response 3: We completely agree with your point. We have implemented the following changes in the revised manuscript and added references 4, 5, 10, 11, 20, etc. to the paper.
Point 4: The quality of the figures needs improvement, particularly the clarity of labels used in the figures.
Response 4: We completely agree with your point. We have improved the quality of the images to enhance their readability and make them more visually appealing.
Point 5: Authors should establish a concrete correlation between the proposed metaheuristic algorithms and the context of path planning. For instance, explain how the proposed methods can be applied to "Global Path Planning," "Local Path Planning," "Obstacle Avoidance," "Re-Planning," etc.
Response 5: We completely agree with your point. According to Figure 4, we have revised the metaheuristic algorithm's principles for path planning, including global path planning, local path planning, and obstacle avoidance. Please refer to Chapter 3, lines 189 to 210, for detailed information.
Point 6: Authors should thoroughly discuss the computational complexity and practical applicability of the proposed methods in path planning applications.
Response 6: The topic of this paper is to synthesize the improvement ideas of intelligent algorithms in recent years for path planning of mobile robots, for problem 6, I am sorry that I may not be able to give detailed and complete application data, but I can provide you with certain information about the hardware devices. Please refer to the relevant content in Table 7 for specifics.
Point 7: Authors should critically discuss different path planning/optimization algorithms, preferably in the form of a table (e.g., Table 1), including additional attributes for comparison such as Advantages, Limitations, Versatility, etc. Gain insights from the following reference: https://doi.org/10.1016/j.robot.2019.103382
Response 7: We completely agree with your point. Please refer to the relevant content in Table 2 for specifics.
Point 8: Authors should provide a brief pseudocode of the main proposed algorithms used in this paper. Gain insights from the following reference: https://doi.org/10.1016/j.eswa.2023.120254
Response 8: We completely agree with your point. Please refer to the relevant content in Figure 6 and Figure 9 for specifics.
Point 9: The Conclusion section should be revised and written in a standard format. The paper's contributions (in such a format) should be explicitly addressed in the Introduction.
Response 9: We completely agree with your point. We have split the conclusion into two sections: 'Discussion' and 'Conclusion.

Reviewer 2 Report
This study proposed a new classification method for the metaheuristic algorithms which are used to solve the path planning problems for mobile robots. The natural-behavior-based metaheuristic algorithms are reviewed with a focus on the firefly algorithm (FA) and cuckoo search algorithm (CS). The latest progress of the research in FA and CS has been reported in detail, which might provide an instrumental reference for researchers in this field. However, some figures and expressions need to be revised to match the requirement of academic writing.
This paper could be published after major revision.
1) The words in all the figures are too small to read. Please use appropriate font size so that the words in the figures appear with almost the same size as those in the main text.
2) Figure 1 presents four kinds of key technology in mobile robots. The relation between the mobility part and the other parts could be provided to complete the entire structural diagram.
3) Please use tables to compare the differences of the major studies listed in Section 3.1 and Section 3.2, respectively, so that the topics, contributions, and performance improvements could be clearly presented.
4) References should be carefully added.
l For example, to make the whole table consistent, Table 1 should provide references for the FA, CS, DA, WOA, and SSA algorithms.
l In addition, Figure 3 mentioned a lot of mainstream metaheuristic algorithms, which should be followed by their references accordingly.
l References for “a variety of fields” in Line 73 and “numerous studies” in Line 79 are absent.
5) The English expression is deficient, especially in Sections 2.1, 2.2, and 3.3, which might need to be reorganized. The paragraph of Section 2.1 is too long. Please avoid using the full names of the scholars in academic writing when citing their works. Moreover, the use of the mark《》is rare in English.
6) The conclusion part should be made concise and clear. The three aspects summarized in the current conclusion part involve a lot of discussions, which might appear long-winded and repetitive. List only conclusions in this part and put the discussions in other sections.
7) Please polish the manuscript to make it concise and clear. Check and fix the minor mistakes like the title format of Section 3 during the revision process
This manuscript needs to be polished to make it concise and clear.
Author Response
Response to Reviewer 2 Comments
Point 1: The words in all the figures are too small to read. Please use appropriate font size so that the words in the figures appear with almost the same size as those in the main text.
Response 1: We completely agree with your point. We have improved the quality of the images to enhance their readability and make them more visually appealing.
Point 2: Figure 1 presents four kinds of key technology in mobile robots. The relation between the mobility part and the other parts could be provided to complete the entire structural diagram.
Response 2: We completely agree with your point. We have made modifications to Figure 1, adding connections between the mobile component and other parts.
Point 3: Please use tables to compare the differences of the major studies listed in Section 3.1 and Section 3.2, respectively, so that the topics, contributions, and performance improvements could be clearly presented.
Response 3: We completely agree with your point. We have added new tables in Sections 3.1 and 3.2 to provide a detailed analysis and summary of the improvements proposed in the FA and CS algorithms presented in the paper. Please refer to Table 5 and Table 6 for specific details.
Point 4: References should be carefully added. For example, to make the whole table consistent, Table 1 should provide references for the FA, CS, DA, WOA, and SSA algorithms. In addition, Figure 3 mentioned a lot of mainstream metaheuristic algorithms, which should be followed by their references accordingly. References for “a variety of fields in Line 73 and “numerous studies” in Line 79 are absent.
Response 4: We completely agree with your point. We have added the corresponding references[25-33],[38-53] in the paper.
Point 5: The English expression is deficient, especially in Sections 2.1, 2.2, and 3.3, which might need to be reorganized. The paragraph of Section 2.1 is too long. Please avoid using the full names of the scholars in academic writing when citing their works. Moreover, the use of the mark《》is rare in English.
Response 5: We completely agree with your point. We have made the necessary changes based on the feedback provided for Sections 2.1, 2.2, and 3.3.
Point 6: The conclusion part should be made concise and clear. The three aspects summarized in the current conclusion part involve a lot of discussions, which might appear long-winded and repetitive. List only conclusions in this part and put the discussions in other sections.
Response 6: We completely agree with your point. We have made revisions to the discussion section in the fourth paragraph. The fourth section contains a substantial amount of discussion, while the fifth section provides a conclusion.
Point 7: Please polish the manuscript to make it concise and clear. Check and fix minor mistakes like the title format of Section 3 during the revision process.
Response 7: We completely agree with your point. We have addressed the feedback and made the necessary improvements to the content of the paper.

Reviewer 3 Report
In this article, the authors have reviewed recent progress in mobile robot path planning driven by the development and improvement of various nature-inspired metaheuristic algorithms. They mostly focus their discussion on the firefly and cuckoo search algorithms, but also discuss other algorithms. This review article is quite timely because of the recent advances that these methods have witnessed.
Overall, the manuscript is relatively well written and easy to follow, despite the fact that some English editing is required. This review is comprehensive and reflects the multitude of approaches that have been recently proposed with respect to nature-inspired metaheuristic algorithms.
I believe the manuscript could be improved, based on the following, relatively minor suggestions:
1) The new classification of metaheuristic algorithms should be mentioned at the end of the Introduction. The authors should keep in mind that one of the roles of the Introduction is to provide an overview of the rest of the manuscript.
2) The authors should increase the resolution and font size of all figures, for better readability.
3) The sentence at lines 166-167, which introduces past classifications for metaheuristic algorithms, is totally unclear. It should be expanded, and examples should be provided to illustrate these classifications.
4) Who is behind "our" at line 489?
5) There is a typo in "several rid algorithms" at line 503.
6) The sentence at line 614 is unfinished.
7) The techniques that the authors mention at lines 626 are not example of General Artificial Intelligence (GAI), but of "regular" Artificial Intelligence (AI). Indeed, GAI relates to more high-level tasks and intelligence, closer to what characterizes human intelligence.
8) At line 653, "so that" should be replaced by "for".
9) References should be provided for all algorithm types in Table 1.
10) Reference [62] does not appear to be an article on BFA. It seems to be an article from the 7th International Symposium on Mechatronics and Industrial Informatics (ISMII), 2021, by Hanhua Cao; Huanping Zhang; Zhendan Liu; Yuhuai Zhou; and Yujuan Wang
11) Not all references in the list [63–71] seem to involve an artificial bee colony algorithm.
12) There are other issues with references, which the authors should carefully check. For example, references [39], [43] and [47] correspond to the same article. Also, references [49] and [72] correspond to the same article.
The manuscript requires moderate English editing: there are only few typos, but many sentences are grammatically incorrect.
Author Response
Response to Reviewer 3 Comments
Point 1: The new classification of metaheuristic algorithms should be mentioned at the end of the Introduction. The authors should keep in mind that one of the roles of the Introduction is to provide an overview of the rest of the manuscript.
Response 1: We completely agree with your point. We have made the following improvements to lines 90-97 in the Introduction section.
Point 2: The authors should increase the resolution and font size of all figures, for better readability.
Response 2: We completely agree with your point. We have improved the quality of the images to enhance their readability and make them more visually appealing.
Point 3:The sentence at lines 166-167, which introduces past classifications for metaheuristic algorithms, is totally unclear. It should be expanded, and examples should be provided to illustrate these classifications.
Response 3: We completely agree with your point. In lines 154 and 160 of this paper, we have added descriptions of the classification methods used in previous algorithms. We have made the necessary modifications based on your feedback.
Point 4: Who is behind "our" at line 489?
Response 4: This was a minor mistake during the translation process, and we apologize for the oversight in our writing. We sincerely appreciate your thorough review and attention to detail. Thank you for your valuable input. We have made the necessary modifications based on your feedback. We have made the necessary modifications based on your feedback.
Point 5: There is a typo in "several rid algorithms" at line 503.
Response 5: This was a minor mistake during the translation process, and we apologize for the oversight in our writing. We sincerely appreciate your thorough review and attention to detail. Thank you for your valuable input. We have made the necessary modifications based on your feedback. We have made the necessary modifications based on your feedback.
Point 6: The sentence at line 614 is unfinished.
Response 6: This was a minor mistake during the translation process, and we apologize for the oversight in our writing. We sincerely appreciate your thorough review and attention to detail. Thank you for your valuable input. We have made the necessary modifications based on your feedback.
Point 7: The techniques that the authors mention at lines 626 are not example of General Artificial Intelligence (GAI), but of "regular" Artificial Intelligence (AI). Indeed, GAI relates to more high-level tasks and intelligence, closer to what characterizes human intelligence.
Response 7: We completely agree with your point. We have made the necessary modifications based on your feedback.
Point 8: At line 653, "so that" should be replaced by "for"
Response 8: This was a minor mistake during the translation process, and we apologize for the oversight in our writing. We sincerely appreciate your thorough review and attention to detail. Thank you for your valuable input. We have made the necessary modifications based on your feedback. We have made the necessary modifications based on your feedback.
Point 9: References should be provided for all algorithm types in Table 1.
Response 9: We completely agree with your point. We have made the necessary modifications based on your feedback.
Point 10: Reference [62] does not appear to be an article on BFA. It seems to be an article from the 7th International Symposium on Mechatronics and Industrial Informatics (ISMII), 2021, by Hanhua Cao; Huanping Zhang; Zhendan Liu; Yuhuai Zhou; and Yujuan Wang
Response 10: We completely agree with your point. We have made the necessary modifications based on your feedback.
Point 11: Not all references in the list [63–71] seem to involve an artificial bee colony algorithm
Response 11: We completely agree with your point. We have made the necessary modifications based on your feedback.
Point 12: There are other issues with references, which the authors should carefully check. For example, references [39], [43] and [47] correspond to the same article. Also, references [49] and [72] correspond to the same article.
Response 12: We completely agree with your point. We have made the necessary modifications based on your feedback.

Reviewer 4 Report
Metaheuristic algorithms are widely used in various optimization problems. This paper presents an overview of nature-inspired metaheuristic algorithms with a particular emphasis on FA and CS in the field of mobile robots path planning. The list of references is quite sufficient. It contains 126 contemporary publications from various specialized journals.
The research topic is relevant. In this paper, the evolution and development trends of metaheuristic algorithms are well presented. It is shown that the best performance is provided by combining different approaches and using hybrid algorithms.
Some suggestions for improving this article.
1. Figure 3 shows the classification chart of metaheuristic algorithms. It is necessary to explain what are the advantages and disadvantages of the three selected groups of algorithms in mobile robots path planning tasks.
2. In the introduction, three groups of mobile robots are distinguished depending on the operating conditions. Here, a more appropriate classification feature is the environment of movement. Marine robots are usually distinguished, which include underwater and surface vehicles. Figure 1 shows the structural diagram of the key technology in mobile robots. Figure 1 shows the schematic diagram of mobile robot path-planning problems. Table 1 summarizes the main application environments. There is no other information about the features and classification of mobile robots path planning tasks. In our opinion, the material is presented one-sidedly. Articles are listed in which the evolution of nature-inspired metaheuristic algorithms is presented with indication of applications. But there is no systematization of these applications. I would like to see a table that presents in more detail the main planning tasks for various mobile robots in various environments with recommendations for applying FA and CS modifications.
3. The question of the implementation of metaheuristic algorithms is not fully considered. I would like to see a table that lists the necessary computing resources and technologies for the FA and CS modifications, computational complexity, computation time, requirements for the on-board computer, etc.
4. The inscriptions on fig. 1. 2, 5, 6 must be enlarged, they are hard to read.
Author Response
Response to Reviewer 4 Comments
Point 1: Figure 3 shows the classification chart of metaheuristic algorithms. It is necessary to explain what are the advantages and disadvantages of the three selected groups of algorithms in mobile robots path planning tasks.
Response 1: We completely agree with your point. We have made the necessary modifications based on your feedback.
Point 2: In the introduction, three groups of mobile robots are distinguished depending on the operating conditions. Here, a more appropriate classification feature is the environment of movement. Marine robots are usually distinguished, which include underwater and surface vehicles. Figure 1 shows the structural diagram of the key technology in mobile robots. Figure 1 shows the schematic diagram of mobile robot path-planning problems. Table 1 summarizes the main application environments. There is no other information about the features and classification of mobile robots path planning tasks. In our opinion, the material is presented one-sidedly. Articles are listed in which the evolution of nature-inspired metaheuristic algorithms is presented with indication of applications. But there is no systematization of these applications. I would like to see a table that presents in more detail the main planning tasks for various mobile robots in various environments with recommendations for applying FA and CS modifications.
Response 2: Regarding this feedback, we have made the following modifications: We have added three new tables, Table 1, Table 5, and Table 6, to the paper. These tables provide a clear classification of mobile robots based on their motion environment, along with corresponding explanations. Additionally, in Sections 3.1 and 3.2, we have included tables that outline the improvement methods of the FA algorithm and CS algorithm for mobile robot path planning. These tables also describe the problems they address.
Point 3: The question of the implementation of metaheuristic algorithms is not fully considered. I would like to see a table that lists the necessary computing resources and technologies for the FA and CS modifications, computational complexity, computation time, requirements for the on-board computer, etc.
Response 3: The topic of this paper is to synthesize the improvement ideas of intelligent algorithms in recent years for path planning of mobile robots, for problem 3, I am sorry that I may not be able to give detailed and complete application data, but I can provide you with certain information about the hardware devices. We have added Table 7 in Section 4.
Point 4: The inscriptions on fig. 1. 2, 5, 6 must be enlarged, they are hard to read.
Response 4: We completely agree with your point. We have improved the quality of the images to enhance their readability and make them more visually appealing.
Round 2
Reviewer 1 Report
The authors have properly addressed the comments and revised the paper accordingly. This version is acceptable for publication.
Acceptable
Reviewer 2 Report
I am satisfied with the revisions provided to the concerns raised by me.
Thank you for your efforts.